# Exercise Training in Non-Hospitalized Patients with Post-COVID-19 Syndrome—A Narrative Review

**DOI:** 10.3390/healthcare11162277

**Published:** 2023-08-12

**Authors:** Johanna Sick, Daniel König

**Affiliations:** 1Department of Sport Science, Centre for Sports Science and University Sports, University of Vienna, 1150 Vienna, Austria; daniel.koenig@univie.ac.at; 2Vienna Doctoral School of Pharmaceutical, Nutritional and Sport Sciences, University of Vienna, 1090 Vienna, Austria; 3Department of Nutritional Sciences, Faculty of Life Sciences, University of Vienna, 1090 Vienna, Austria

**Keywords:** post-acute COVID-19 syndrome, long COVID, rehabilitation, exercise therapy

## Abstract

Post COVID-19 Syndrome (PCS) is the persistence of symptoms after an infection with SARS-CoV-2 in both hospitalized and non-hospitalized COVID-19 survivors. Exercise was proposed as a rehabilitation measure for PCS and early studies focused on patients post-hospital discharge. The objective of this review is to summarize the results of trials investigating exercise interventions in non-hospitalized subjects with PCS and propose practical recommendations concerning safe exercise programming. A literature search in the databases MEDLINE and Scopus was conducted until 26 July 2023 and resulted in seven studies that met the criteria. In total, 935 subjects with PCS were investigated. Exercise enhanced aerobic fitness and physical function and relieved symptoms of dyspnea, fatigue and depression. Participants reported lower Post COVID-19 Functional Status scores post-intervention. The exercise programs were well tolerated with no adverse events. To ensure safety, medical examinations should take place in advance and there should be a regular evaluation of the individual responses to the training. Caution is advised when working with patients suffering from post-exertional malaise or diagnosed with myalgic encephalomyelitis/chronic fatigue syndrome. There is a growing need for additional randomized controlled trials to investigate the effectiveness and safety of exercise in individuals with PCS.

## 1. Introduction

Up to this date, 16 June 2023, 767,000,000 individuals worldwide have been affected by coronavirus disease 2019 (COVID-19), leading to a global healthcare crisis and almost 7 million deaths (WHO Coronavirus Dashboard). While its infectious agent, the severe acute respiratory syndrome coronavirus 2 (SARS-CoV-2), mostly causes asymptomatic or mildly symptomatic acute cases, survivors soon noticed long-lasting sequelae and coined the term “Long-haul COVID” or “Long COVID” [1]. Despite numerous scientific efforts to describe the newly emerged syndrome, there is no uniform definition or terminology. Experts suggest the terms “acute post-COVID” and “ongoing symptomatic COVID-19” for symptoms lasting longer than 4 weeks, and “long post-COVID” or “Post-COVID-19 syndrome” (PCS) after a persistence of 12 weeks [2,3]. These definitions are in line with the COVID-19 rapid guideline of the National Institute for Health and Care Excellence, in which PCS is defined as “signs and symptoms that develop during or after an infection consistent with COVID-19, continue for more than 12 weeks and are not explained by an alternative diagnosis” [4].

Due to its diverse symptomatic manifestations, PCS can be seen as a multi-organ disease affecting various systems: cardiovascular, respiratory, gastrointestinal, musculoskeletal and nervous systems, as well as mental health and others [5]. Several works studied the type, number and prevalence of symptoms [6,7,8,9]. The most commonly observed symptoms were fatigue, dyspnea, myalgia, disturbed taste and/or smell, cognitive impairment, insomnia, anxiety and depression. The latest data reported a pooled prevalence of 52.6% and 34.5% in hospitalized and non-hospitalized patients, respectively [10]. Being of female gender, belonging to an ethnic minority group, smoking, an increased BMI, hospitalization, and the presence of various comorbidities constitute a higher risk for developing PCS [11,12,13].

Long lasting sequelae with similar symptoms are also observed in the sports athlete population, although with a smaller prevalence of 8.3% [14]. Studies report impaired respiratory patterns in those athletes suffering from persistent symptoms [15,16], which may lead to decreased performance parameters and a greater metabolic demand [17]. 

The pathophysiology is still poorly understood. Generally, studies distinguish between symptoms that are either related or unrelated to organ damage or dysfunction caused by acute COVID-19 [18]. Several abnormalities, such as immune dysregulation, microbiota dysbiosis, autoimmunity and immune priming, endothelial dysfunction and dysfunctional neurological signaling were proposed as possible underlying mechanisms [19]. An overlap of these processes may occur and contribute to the heterogenous manifestations of the syndrome [20]. A particular phenotype of PCS shows similarities to myalgic encephalomyelitis or chronic fatigue syndrome (ME/CFS), with fatigue and exercise intolerance (Post Exertional Malaise–PEM) being the most characteristic symptoms [21]. Indeed, SARS-CoV-2 was identified as an infectious trigger for post-viral fatigue and a considerable number of PCS sufferers were diagnosed with ME/CFS [22,23].

The large number of people affected by PCS underlines the importance of effective and safe rehabilitation measures. Exercise, which has shown to be successful in treating similar conditions [24,25], was explored as a non-pharmacological therapy for post-viral syndromes, including PCS [26]. Different authors stated its possible benefits for improving patients’ functional capacity, health-related quality of life (HRQoL), and severity of symptoms [3,27]. So far, multiple studies have investigated the effect of pulmonary rehabilitation (PR) in former hospitalized patients post COVID-19, in both a supervised and telerehabilitation setting [28,29,30]. A meta-analysis done by Chen and colleagues [31] showed an increase in exercise capacity and HRQoL after PR. Overall, exercise training seems to be a successful and feasible rehabilitation strategy for individuals suffering from long-lasting symptoms as a result of an infection with SARS-CoV-2 [32]. 

However, it should be noted that a vast majority of studies investigated patients post hospital discharge, often a relatively short time after the acute phase of COVID-19. A prolonged immobility due to bed confinement or ICU admission could contribute to the physical consequences of the disease, such as decreased exercise capacity, impaired lung function and persistent fatigue [33,34]. We need more information that explores the efficacy of exercise interventions in home-confined subjects, which apply the most common definition of PCS (>12 weeks post-infection). We therefore aimed to conduct a narrative review in order to further investigate the effects of exercise training and its possible benefits in the rehabilitation of non-hospitalized patients with PCS. Furthermore, this work provides recommendations for a safe prescription of exercise to this patient population.

## 2. Methods

A comprehensive literature search was conducted using the electronic databases MEDLINE (PubMed) and Scopus between 15 March 2023 and 4 May 2023 in order to identify studies that investigated the efficacy of exercise interventions in patients with PCS. An updated search was performed on 26 July 2023. The following medical subject headings terms were used: Post Acute COVID-19 Syndrome, Post-COVID Condition, Long COVID, Long Haul COVID-19, exercise, aerobic exercise, anaerobic exercise, exercise therapy, endurance training, resistance training, strength training, physical activity, rehabilitation, HRQOL, fatigue and dyspnea. The inclusion criteria were as follows: prospective peer-reviewed trials examining exercise as a therapeutic measure in subjects with PCS (>12 weeks post-infection with SARS-CoV-2) that report outcomes (primary or secondary) related to the symptomatology, functional or exercise capacity, and HRQoL, were written in English, and published from 2020 onwards. Articles that did not meet these criteria were excluded. After the removal of duplicates, two authors independently screened the literature using titles and abstracts, then full texts were reviewed for eligibility. The following data were extracted and synthesized in a table using standardized data extraction methods: first author, year of publication, type of study, characteristics of subjects and interventions, exercise protocols, outcome measures and results. 

## 3. Results

### 3.1. Characteristics and Participants of Included Studies

The literature search resulted in a total of seven studies that met the inclusion criteria. In summary, 935 adult participants (672 females and 263 males) with a combined mean age of 48.0 years were investigated. The time of inclusion ranged from a mean of 4.4 months [35] to up to 12 months post-SARS-CoV-2 infection [36]. Two trials included only subjects with PCS that were not hospitalized due to COVID-19 [37,38], and four examined both hospitalized and non-hospitalized (mild) patients; however, each with a majority of 90.6% [39], 60.8% [36], 62.0% [35] and 86.7% [40] of mild COVID-19 cases. The last study, which comprised a sample of health care workers post COVID-19, did not provide any information about the hospitalization status [41]. Instead, the participants were categorized according to their Post-COVID-19 Functional Status (PCFS) and allocated to the mild symptom group (PCFS 0 and 1; 35.7%) or the severe symptom group (PCFS 2 and higher, 64.3%). Among the studies were two randomized controlled trials (RCT), one comparing concurrent training with (CTRM) and without (CT) inspiratory muscle training to self-management rehabilitation recommendations and inspiratory muscle training alone (RM) [38], as well as one comparing an exercise group (EX) to self-management rehabilitation recommendations [37]. Furthermore, we included the results of an intervention trial with two parallel groups [41], three prospective trials evaluating outpatient rehabilitation programs [35,36,40] and a quasi-experimental clinical trial with digital physiotherapy [39]. The study characteristics and outcomes are displayed in Table 1.

### 3.2. Exercise Interventions

Table 2 provides an overview of the exercise interventions. One trial conducted a supervised concurrent training that combined two weekly sessions of resistance and moderate-intensity variable training and one weekly session of light-intensity continuous training [37]. In a second study the same protocol was combined with inspiratory muscle training in one of the parallel groups [38]. Hasenoehrl and colleagues [41] used a low-intensity, high-repetition resistance exercise method in a supervised manner. Both pulmonary rehabilitation programs [35,36] included three weekly sessions of aerobic, resistance and breathing exercises as well as additional patient education and psychological counselling. No information on exercise intensity could be found. The digital physiotherapy intervention by Estebanez-Pérez and colleagues [39] implemented a personalized four-week program that followed individual assessments of patients. The rehabilitation program of Smith et al. [40] was group-based and split into two phases, a virtual and a face-to-face intervention consisting of supervised and unsupervised sessions. 

### 3.3. Physical Function

Physical function was assessed via various forms of the Sit-to-Stand Test (STS) in all but one study, submaximal [37,38] and maximal cardiopulmonary exercise testing (CPET) [36,41] or a maximal exertion test on a cycle ergometer [35], the 6 Minute Walking Test (6MWT) [35,36,41], and the Short Performance Physical Battery Test (SPPB) [36,39]. Additionally, muscular strength was tested via handgrip, isometric knee extension and progressive submaximal and maximal loading tests in two studies [37,38]. 

All studies reported significant improvements of STS. Estimated maximal oxygen consumption (VO_2_max) significantly increased by 2.1 mL/kg/min in EX [37] as well as 2.9 mL/kg/min in CT and 2.5 mL/kg/min in CTRM [38] with no changes in the control groups of the RCTs. Peak oxygen consumption (VO_2_peak) improved by 2.4 mL/kg/min in the severe symptom group of Hasenoehrl et al. [41]. Nopp et al. [35] observed a significant improvement in maximal workload of 21.8 W after 6 weeks of pulmonary rehabilitation. 6MWT significantly increased by 68.9 m and 57.6 m in the severe and mild symptom group, respectively, [41] and by 62.9 m as well as 62.5 m, as observed after the PR programs [35,36]. Participants reached significantly higher SPPB scores after 6 weeks of PR (+2 points) [36] and 4 weeks of digital physiotherapy (+1.22 points) [39]. Significant improvements in muscular strength could be measured by progressive submaximal and maximal loading tests in bench press and half squats in all of the exercise groups [37,38].

In summary, exercise and multidisciplinary rehabilitation interventions enhanced physical function in patients with PCS. Estimated VO_2_max and VO_2_peak increased in three out of four trials, most likely because one study conducted a 6-week intervention [36] while the other trials that conducted CPET lasted 8 weeks each. These improvements can be seen as clinically relevant, as aerobic fitness is a strong predictor of mortality and positively correlates with health-related quality of life [42,43]. Furthermore, consistent positive results were observed for 6MWT, STS and SPPB, which reflect significant enhancements in functional capacity. In the study of Smith et al. [40], significant increases in the Duke Activity Status (DASI) were observed, which indicates improvements in functional status. As reported in the RCTs, certain improvements also occurred without supervised training and with the use of inspiratory muscle training alone; however, this does not apply to exercise capacity.

### 3.4. Symptoms and Patient Reported Outcomes 

All but one study assessed various patient reported outcomes (PROs) regarding PCS symptoms and quality of life. Most commonly used was the modified Medical Research Council Dyspnea scale (mMRC) [35,36,37,38], which measures perception of dyspnea, and the PCFS scale quantifying functional limitations post COVID-19 [35,37,38,41]. Fatigue was assessed via the Chalder Fatigue Scale (CFS) and the Fatigue Severity Scale (FSS) [37,38], the Fatigue Assessment Scale (FAS) [35], the Brief Fatigue Inventory (BFI) [41] and the Modified Fatigue Impact Scale (MFIS) [36]. Jimeno-Almazán et al. [37] additionally used the Short Form DePaul Symptom Questionnaire (DSQ-14) to screen for ME/CFS symptomatology. Furthermore, three studies evaluated symptoms of depression (Patient Health Questionnaire 9—PHQ-9) and anxiety (Generalized Anxiety Disorder 7—GAD-7), as well as resilience (Brief Resilience Scale) and stress (Perceived Stress Scale 10) [37,38,41]. In addition, Smith et al. [40] assessed mental well-being via The World Health Organization- Five Well-Being Index (WHO-5) as well as breathlessness using the Dyspnea-12 tool (D-12). HRQoL was measured via the 12-item Short Form Survey (SF-12) [37,38] and the EuroQol Group five-dimension five-level questionnaire (EQ-5D-5L) [35].

Dyspnea and functional status improved significantly in all studies that used the corresponding scales (mMRC, D-12 and PCFS). While the mMRC score decreased similarly in both the exercise and the control group in one RCT (by 0.88 and 0.56, respectively) [37], a significant pre–post difference was only found when exercise was combined with inspiratory muscle training in another trial [38]. A group effect in favor of exercise was observed for PCFS with a decrease of 1.5 in one study [37]. Furthermore, CT resulted in a significant pre–post difference in the number of participants with a PCFS score < 2 (from 3 to 14) [38]. Improvements of perceived fatigue occurred in multiple studies, as assessed by CFS bimodal (from 8.1 to 3.5) and CFS Likert (from 22.8 to 11.4) [37], FSS (from 5.0 to 3.4) [37], FAS (from 26 to 20) [35] and MFIS (from 37 to 27) [36]. Additionally, one study [38] found that significantly more participants had a CFS Likert score < 18 and an FSS score < 4 in CT (from 5 to 17 and 6 to 14, respectively) and CTRM (from 8 to 15 and 2 to 8, respectively). In both RCTs, significant group effects favoring the exercise groups were found for all of the fatigue measures. In multiple studies, improvements in anxiety, depression and other mental well-being measures were observed and presented in Table 2. 

For HRQoL, significant improvements were reported for the EQ-5D visual analog scale in two studies [35]. An increase in the EQ-5D-5L utility score was furthermore assessed by Smith et al. [40]. Jimeno-Almazán et al. [37] observed a significant group effect in favor of exercise in the physical activity domain of the SF-12 (from 35.7 to 47.8), while pre–post increases without differences between groups were found in the same domain in CT (from 35.2 to 48.2), CTRM (from 33.8 to 41.0) and RM (from 35.8 to 44.1) in another trial [38]. The mental health domain significantly increased in CTRM only (from 39.5 to 44.8). 

The results presented above suggest that exercise improves symptoms of dyspnea, fatigue and depression in patients with PCS, which translates into lower PCFS scores. Since scores of 2 and higher are associated with functional impairments in work and usual activities, the average improvements in the reviewed trials can be seen as clinically relevant [44]. Positive effects on HRQoL, which were assessed in four trials, were observed in certain domains only. A control group following self-management rehabilitation recommendations improved in mMRC, PCFS, PHQ-9 and GAD-7 after 8 weeks [37]. The same practice, however, did not lead to changes in PROs in the second RCT [38].

### 3.5. Safety 

Preceding the trials, physical examinations and screenings for medical history and contraindications for exercise training were administered. Common procedures included electrocardiogram and echocardiogram, pulmonary function tests, CPET and laboratory blood testing. Two publications did not provide information regarding on-site clinical assessments [39]. 

No adverse events were reported during the course of the interventions in any of the studies. Information about the number of dropouts was provided by Nopp et al. [35] (*n* = 6), Estebanez-Pérez et al. [39] (*n* = 4), Jimeno-Almazán et al. [37] (*n* = 1) and Jimeno-Almazán et al. [38] (*n* = 3). The reasons for dropouts include adherence problems, personal issues, injury and sickness, SARS-CoV-2 reinfection and fear of reinfection. The authors of one study additionally state that no dropouts occurred due to tolerance issues [37]. However, the authors mention the importance of the management of PEM as well as an individual treatment of participants by adapting the intensity of training sessions. Both multidisciplinary rehabilitation programs as well as the blended community-based rehabilitation also used an individualized approach and based their programs on detailed health assessments of PCS patients [35,36]. Smith et al. [40] prescribed an intervention according to the subjects’ physical fitness and functional capacity. Lastly, the exercise intervention of Hasenoehrl et al. [41] and the digital physiotherapy intervention of Estebanez-Pérez et al. [39] allowed for personalized adjustments of intensity and exercise selection.

Some, but not all, studies reported the use of monitoring instruments during or after training sessions. The most common tools were scales of subjective perceived exhaustion [37,38,40,41] and heart rate monitors [37,38,41]. The measurement of blood oxygen saturation was mentioned in one study [35]. Furthermore, in a majority of the studies, exercise sessions were supervised by healthcare professionals or certified sport scientists [35,37,38,39]. Hasenoehrl et al. [41] and Smith et al. [40] implemented both supervised and unsupervised sessions. 

In the reviewed studies, medical examinations were carried out before the exercise interventions and all of the training regimes were individually adapted, if needed. These precautions, as well as the use of rating of perceived exertion (RPE) scales, heart rate monitoring and the supervision of most sessions likely contributed to the absence of exercise tolerance issues and adverse events. Despite these findings, concerns regarding the worsening of symptoms in response to exercise should not be dismissed. Sessions should ideally only take place in an ambulatory setting under the supervision of qualified training personnel and be regularly evaluated, as highlighted by Jimeno-Almazán et al. [38]. 

## 4. Practical Recommendations and Discussion

### 4.1. Medical Examinations

Before starting an exercise program after an infection with SARS-CoV-2, medical examinations have been recommended by most authors [45]. However, most recently, a pragmatic approach that is governed by initial symptom burden and resolution of symptoms has been proposed, which does not generally recommend a medical investigation before returning to physical activity. A similar conclusion was made regarding athletes returning to sports after a mild SARS-CoV-2 infection. The authors state that an extensive cardiorespiratory screening is not necessary; however, more research is needed in that area [46]. In any case, a medical practitioner should be consulted if symptoms worsen or if there are repeated adverse responses to exercise such as fatigue and exercise intolerance [47]. Cattadori et al. [48] further suggested that post-COVID-19 exercise protocols should not be administered to patients with various contraindications, including a resting heartrate above 100 bpm, abnormal blood pressure or blood oxygen saturation, temperature fluctuation, ongoing respiratory symptoms and fatigue that is not relieved by rest. Despite these recommendations, the authors mentioned that there is currently a lack of evidence-based strategies and that most of the proposals concerning exercise and post-COVID-19 are retrieved from expert consensus statements.

A key diagnostic tool for the prescription of exercise is the use of CPET. It helps to identify the potential causes for reduced physical capacity in PCS patients and makes it possible to specifically tailor the program to the individual [34]. PEM assessment after CPET could further aid in identifying patients with exercise intolerance and in deciding whether an exercise program should be initiated or not [49]. 

In light of the considerations above, a prior medical examination would be advisable for safety reasons. Apart from the fact that contraindications for physical training as proposed by the American College of Sports Medicine [50] are recognized, the examination should also focus on exercise intolerance or fatigue. In addition, CPET is recommended prior to the prescription of an exercise program. 

### 4.2. Exercise Intolerance and ME/CFS

While many of the expert opinions emphasize the potential advantages of physical training for individuals with PCS, a recent controversy concerning its possible risks has raised attention [51]. The debate centers around individuals who are affected by PEM or have been diagnosed with ME/CFS after the infection with SARS-CoV-2. In fact, relapses triggered by exercise have been identified as a frequent symptom in PCS cases [52]. A study by Jason and Dorri [22] found that 58% of participants with Long COVID meet the criteria for ME/CFS, and similar results were published by Twomey and colleagues [53], who reported the presence of chronic fatigue and PEM in a majority of subjects. The authors stated that exercise may be beneficial for some, but not all, PCS patients. Furthermore, they advocate for individualized multidisciplinary rehabilitation programs and underscore the importance of the reporting of symptom exacerbation and adverse events.

Before prescribing a training program to PCS patients, a thorough screening for exercise intolerance in form of PEM [53] is recommended. The DePaul Symptom Questionnaire–Post-Exertional Malaise (DSQ-PEM) can be used as a corresponding diagnostic tool [54]. The questionnaire exhibits good utility in clinical assessments. ME/CFS patients can be differentiated from healthy subjects with a cutoff score of 20 on the PEM subscale [55]. Patients affected by PEM or who were diagnosed with ME/CFS should not be exposed to conventional exercise programs, as it is potentially harmful for this population [56]. A pacing protocol with incremental phases of physical activity according to RPE scores could be a beneficial alternative [57]. 

### 4.3. Rehabilitation Approaches

A frequent approach to the post-care of patients with PCS has been the administration of PR, which commonly comprises exercise, educational and behavioral programs along with medical and psychological counselling [58]. PR has recently been evaluated as a therapeutic measure in both patients with acute COVID-19 and those with post-COVID-19 conditions. This seems to be an effective and safe practice [31,59]. Such implications can also be drawn from other forms of rehabilitation such as cardiac rehabilitation or the treatment of cognitive impairments and various brain disorders, all of which make use of the beneficial effects of physical training [60,61,62]. While aerobic exercise, often accompanied by resistance training, is at the core of pulmonary and cardiac rehabilitation [58,60], so called mind–body exercises were found to be advantageous in improving cognitive function. They combine slow, coordinated movements with relaxation and body awareness techniques and are of low to moderate intensity [61].

PCS patients, who can be affected by a wide variety of symptoms, might benefit from a multicomponent program that implements elements of all rehabilitation approaches described above [27]. Aerobic exercise could increase cardiac output and mitochondrial function, which are both hypothesized to be impaired in patients after a COVID-19 infection resulting in a reduced oxygen uptake [34]. Functionality and muscular strength might be recovered by resistance training, while mind–body exercises could furthermore be recommended to subjects with a neuropsychological or cognitive symptom burden [61]. This practice could also enhance autonomic function, which is dysregulated in some subjects post COVID-19 [63]. Finally, inspiratory muscle training can be considered as an additional tool in the recovery of PCS with dyspnea, as it was shown to improve symptoms of breathlessness in a randomized controlled trial of McNarry et al. [64]. Breathing exercises were part of both reviewed pulmonary rehabilitation programs [35,36] and were carried out along with aerobic and resistance training in one of the included RCTs [38]. However, it was not advantageous when compared to concurrent training and inspiratory muscle training alone. These findings are consistent with those of the literature reviewing PR in COPD patients [65]. The following section aims to provide more specific recommendations for the implication of two key components of rehabilitation programs, endurance and resistance exercise. It is important to recognize, however, that a multifaceted approach should also include education and, if needed, medical and psychological counselling. 

### 4.4. Exercise Programs

In the evaluated studies, endurance exercise was of low to moderate intensity. Both objective (heartrate) [37,38,41] and subjective (RPE) measures [37,38,40,41] were used to govern maximal intensity during the sessions in some, but not all studies. Hasenoehrl et al. [41], who did not supervise the endurance training in an ambulatory setting, advised their participants to train at the first ventilatory threshold (VT1). This approach can be recommended for PCS patients, since it appears to be safe and feasible and has the ability to improve impaired VO_2_max as well as fatty acid metabolism [66,67]. The intensity and volume of endurance exercise can be increased in accordance with individual responses to the training. For instance, submaximal intervals and prolonged sessions at VT1 may be added to further enhance aerobic and anaerobic capacity [68]. Because of possible relapses and the deterioration of PCS symptoms, maximal exertion should be avoided [69,70]. We further suggest an ongoing evaluation of symptom severity and other relevant health markers such as sleep quality, energy level and general mood during the course of the intervention. 

Due to its potent effects in building muscular strength, improving functional status and enhancing metabolic health, aerobic training should be accompanied by resistance exercise [71]. The approaches regarding the mode and intensity of resistance training differed between the reviewed studies. While Jimeno-Almazán et al. [37] employed the one-repetition maximum (50%) in combination with velocity-based techniques to determine the training loads, only the range from 8 to 12 repetitions was instructed in the digital physiotherapy trial [39]. Hasenoehrl et al. [41] used timed intervals of 30–60s as well as an RPE score (1–10) of 7–8 during the initial 2 weeks, and 9–10 for the remaining intervention. The participants were thus encouraged to reach muscular failure. It is worth mentioning that comparable adaptations can also be achieved when repetitions are not executed to exhaustion [72], which should be considered for safety and tolerance reasons in this particular population. Therefore, there is a rationale to propose that resistance training in PCS patients should be of submaximal intensity. Rather than using objective measures such as the one-repetition maximum, load-management strategies like the repetitions in reserve or autoregulation could also be of advantage [73,74]. In this manner, individual responses to the training stimuli and the fluctuation of symptom severity can be taken into account. A summary of the safety considerations and practical training recommendations are displayed in Figure 1.

### 4.5. Limitations 

This work, along with its included studies, presents some limitations. Firstly, only two trials solely involved non-hospitalized participants [37,38], and one contained no information about the hospitalization status [41]. Even though the majority of remaining subjects were home-confined during their infection with SARS-CoV-2, the inclusion of patients requiring in-patient treatment blunts the specificity of the results. Furthermore, in those studies [35,36,39,40], no information about interventions such as physiotherapy during the hospital stay was provided. 

Secondly, a risk of bias and quality assessment of the included studies was not performed. However, it can be assumed that the risk of bias is high and the overall quality of trials is low. This is due to a lack of randomized controlled trials and some studies having an observational or quasi-experimental design [35,36,39], along with a large heterogeneity of reported outcome measures and a small number of only seven included studies. 

Lastly, there was a broad heterogeneity of exercise interventions and, in part, missing information about the exact exercise selection, frequency and intensity. Even though all of the studies employed a combination of some type of resistance and endurance exercises, these varied greatly in their specific modality (for instance, individual physiotherapy [39], group-based bodyweight or resistance band exercises [40,41], machine- and weight-based training [37,38]) as well as overall duration (4–12 weeks). Since varying training modalities have different effects and thus evoke different responses in the individual subject groups, it is difficult to compare their efficacy. Furthermore, some trials included additional, unsupervised activities [39,40,41] or did not provide information about the specific type and intensity of endurance and resistance exercise [35,36]. 

## 5. Conclusions

The results from the studies included showed that exercise improved physical function, symptom severity and overall functional status in patients with PCS, the majority of whom were not hospitalized due to acute COVID-19. The most consistent positive results were observed for Sit-to-Stand and 6 Minute Walking Tests in the physical function domain, and for fatigue, dyspnea and depression in the patient-reported outcomes domain. The improvements further translated into better Post COVID-19 Functional Status scores. Subjects seemingly showed good tolerance to the interventions, since no dropouts or adverse events due to negative responses to exercise occurred. However, due to the missing information about dropouts or adherence rates in some of the reviewed studies, this finding should be interpreted with caution. Before administering training programs to PCS patients, a number of considerations are required. Individuals who suffer from PEM or were diagnosed with ME/CFS might experience a deterioration of their state in response to exertion. Therefore, it is important to identify and exclude contraindications through a prior medical examination. The exercise interventions should follow a slow, progressive approach and be regularly evaluated in regard to symptom severity and the health status of patients. 

It is important to note that the presented evidence inevitably suffers from the large heterogeneity of studies and their overall high risk of bias. Thus, future research on exercise in PCS should follow a more targeted approach regarding the type, intensity and duration of interventions, as well as the type of studies. This could imply using precise tools of exercise prescription such as heart rate or RPE-based intensity zones, supervised interventions, precise reporting of adherence, dropouts and adverse events, and, lastly, randomized controlled designs. Until a greater amount of evidence about this relatively new condition and its rehabilitation has emerged, this and similar works provide preliminary recommendations about the safety and employment of exercise in individuals with PCS. 

## Figures and Tables

**Figure 1 healthcare-11-02277-f001:**
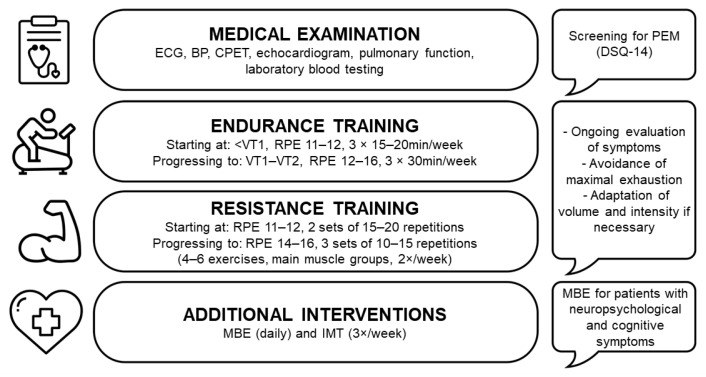
Safety considerations and practical recommendations for exercise training in PCS patients. Electrocardiogram (ECG), blood pressure (BP), ventilatory threshold 1 and 2 (VT1 and VT2), resistance training (RT), mind–body exercises (MBE), inspiratory muscle training (IMT).

**Table 1 healthcare-11-02277-t001:** Summary of studies.

Author	Type of Study	Subjects	Physical Function Outcomes	Patient Reported Outcomes	Conclusion of Authors
Nopp et al. [35]	Prospective observational cohort study	*n* = 58 (25 females); 46.8 ± 12.6 years; 4.4 ± 2.0 months post COVID-19.; 62.0% non-hospitalized; among hospitalized: average length of stay 19.6 days, 11 severe and 11 critical cases	Maximal workload ↑ 6MWT ↑ STS ↑	mMRC ↓ PCFS ↓ FAS ↓ EQ-5D index score ↔ EQ-5D VAS ↑	Significant improvements in exercise capacity, symptoms and quality of life
Ostrowska et al. [36]	Prospective observational single-center study	*n* = 97 (53 females); 60 (50–68) years; 3–12 months post COVID-19; 60.8% non-hospitalized; among hospitalized 24.7% with respiratory failure	VO_2_peak ↔ 6MWT ↑ STS ↑ SPPB ↑	mMRC ↓ MFIS ↓	Significant improvements in symptoms and physical capacity in a majority of patients after rehabilitation
Jimeno-Almazán et al. [37]	RCT (CT vs. self-management rehabilitation recommendations)	*n* = 39 (29 females); 45.2 ± 9.5 years; 33 ± 20.5 weeks post COVID-19; non-hospitalized	estimated VO_2_max ↑* (EX) STS ↑* (EX & C) BP ↑* (EX) HSQ ↑* (EX & C) Handgrip ↔ Leg extension ↔	mMRC ↓ (EX & C) ↔* PCFS ↓* (EX & C) CFQ-11 ↓* (EX) FSS ↓* (EX) SF-12 PA ↑* (EX) MH ↔* GAD-7 ↓ (EX & C) ↔* PHQ-9 ↓* (EX & C) DSQ-14 ↔	Significant improvements in health and performance markers after exercise compared to rehabilitation recommendations
Jimeno-Almazán et al. [38]	RCT (CT vs. RM vs. CTRM vs. self-management rehabilitation recommendations)	*n* = 80 (55 females); 45.3 ± 8.0 years; 39.3 ± 23.3 weeks post COVID-19; non-hospitalized	estimated VO_2_max ↑ (CT & CTRM) ↔* BP ↑* (CT & CTRM) HSQ ↑ (CT & CTRM) ↔* Handgrip ↔	mMRC ↓ (CT & CTRM) ↔* PCFS ↓ (CT) ↔* FSS ↓* CFS ↓* (CT&CTRM) SF-12 PA ↑ (CT, CTRM & RM) ↔* SF-12 MH ↑ (CTRM) ↔* GAD-7 ↓ (CTRM) ↔* PHQ-9 ↓* (CT & CTRM)	Significant improvements in fitness, symptom severity and health status after concurrent training with and without inspiratory muscle training
Estebanez-Pérez et al. [39]	Quasi-experimental pre–post study	*n* = 32 (23 females); 45.9 ± 10.7 years; >12 weeks post COVID; 90.6% non-hospitalized; among hospitalized 2 out of 3 admitted to ICU	STS ↑ SPPB ↑	N/A	Significant improvements in functional capacity and high adherence rate
Smith et al. [40]	Prospective interventional trial	*n* = 601 (465 females); 47.0 ± 10.0 years; 9.8 ± 5.0 months post COVID-19; 86.7% non-hospitalized; among hospitalized average length of stay 10 days, 16.5% admitted to ICU	STS ↑ DASI ↑	D-12 ↓ WHO-5 ↑ EQ-5D-5L ↑ EQ-5D VAS ↑	Significant and clinically meaningful improvements in dyspnea, functional capacity, mental wellbeing and HR-QoL
Hasenoehrl et al. [41]	Intervention trial with two parallel groups (MSG vs. SSG)	*n* = 28 (22 females); 45.8 ± 11.0 years; 6.1 ± 3.1 months post COVID-19; no information about hospitalization	VO_2_peak ↑ (SSG only) 6MWT ↑ STS ↑	PCFS ↓ BFI ↓ GAD-7 ↓ PHQ-9 ↓ PSS-10↓ BRS ↔	Significant improvements of physical fitness and psychological outcomes, higher benefit for SSG

Legend: increase (↑), decrease (↓), no change (↔); (↑* and ↓*) significant group or interaction effect in favor of exercise, (↔*) no group or interaction effect. Abbreviations: bench press (BP), Brief Fatigue Inventory (BFI), Brief Resilience Scale (BRS), control group (C), Duke Activity Status Index (DASI), Dyspnea-12 tool (DS-12), half squat (HSQ), mild symptom group (MSG), not applicable (N/A), Perceived Stress Scale 10 (PSS-10), resistance exercise (RE), severe symptom group (SSG), Visual Analog Scale (VAS), World Health Organization- Five Well-Being Index (WHO-5). Data are presented as mean ± standard deviation.

**Table 2 healthcare-11-02277-t002:** Characteristics of exercise interventions.

Author	Type of Intervention	Duration and Frequency	Type and Intensity of Exercise
Nopp et al. [35]	Multi-professional outpatient rehabilitation	6 weeks; 3×/week	Endurance, strength and inspiratory muscle training according to the Austrian guidelines for outpatient pulmonary rehabilitation
Ostrowska et al. [36]	Multidisciplinary outpatient rehabilitation	6 weeks; 3×/week	Aerobic, resistance and breathing exercises
Jimeno-Almazán et al. [37]	Multicomponentexercise program	8 weeks; 3×/week	Resistance exercises with 3 × 8 repetitions; moderate intensity variable training (70–80%/55–65% HRR); light intensity continuous training (65–70% HRR);
Jimeno-Almazán et al. [38]	Multicomponent exercise program with and without inspiratory muscle training	8 weeks; 3×/week	Resistance exercises with 3 × 8 repetitions; moderate intensity variable training (70–80%/55–65% HRR); light intensity continuous training (65–70% HRR);
Estebanez-Pérez et al. [39]	Personalized digital physiotherapy	4 weeks; 3–5×/week	Progressive strength training (1–3 muscle groups, 8–12 repetitions, load increase by 5–10%/week); additionally recommended: walking, jogging or swimming at low intensities
Smith et al. [40]	Blended community rehabilitation program	12 weeks; 3×/week	Combination of cardiovascular, strength-based, and mobility exercises; intensity and volume prescribed according to participants’ functional capacity
Hasenoehrl et al. [41]	Supervised resistance exercise and unsupervised endurance exercise	8 weeks, 5×/week	Bodyweight and resistance band exercises, 30–60 s repetition maximum (low-intensity high-repetition); aerobic training 3 × 20 min at VT1

Abbreviations: heart rate reserve (HRR), ventilatory threshold 1 (VT1).

## Data Availability

Not applicable.

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
