# Peer review of "Exercise Training in Non-Hospitalized Patients with Post-COVID-19 Syndrome—A Narrative Review"

_healthcare, 2023, doi:10.3390/healthcare11162277_

Round 1
Reviewer 1 Report
This was a study performed to summarize the effectivity and safety of exercise training, including aerobic exercise, resistance training and body-mind exercise, for non-hospitalized patients with post-COVID-19 syndrome (PCS) and propose practical recommendations concerning safe exercise programming. This is indeed a topic of interest given the increased focus on the importance of effective and safe rehabilitation measures in the large number of people affected by PCS. The manuscript suffers from a superficial review of this idea and a total number of 6 studies, especially only randomized controlled trials (RCTs) were met the inclusion criteria. Moreover, the results suggest exercise training enhance aerobic fitness and physical function, relieve the symptoms of dyspnea, fatigue and depress, and then lower Post COVID-19 Functional Status scores. Unfortunately, the small included literatures, the large differences in the modality, intensity, and frequency of exercise intervention, and the heterogeneity of the outcomes dimmish the overall impact. The paper needs very significant improvement.
Strengths:
This study focuses on the effectivity and safety of home exercise rehabilitation for patients with post-COVID 10 syndrome, which has very good research value and social practice significance.
Limitations:
The beneficial effects vary greatly at different modality, amount and frequency of exercise intervention in the patients with post-COVID-19 syndrome, and there are obvious differences in response to exercise intervention. Moreover, too small included researches make the research results unconvincing.
1. What is the time frame for inclusion? From 15 March 2023 to 4 May 2023, is the time of the literature search conducted or included in the publication? Please clarify.
2. Should the sentence “Exercise enhanced …… and symptoms of dyspnea …” be replaced as “Exercise enhanced …, and relieved symptoms of dyspnea…”. Please clarify.
3. It was suggested that the practical recommendations and discussion in Part 4 should correspond to Part 3, i.e., to make corresponding practical recommendations for the results of the study.
Author Response
Replies to Reviewer 1:
This was a study performed to summarize the effectivity and safety of exercise training, including aerobic exercise, resistance training and body-mind exercise, for non-hospitalized patients with post-COVID-19 syndrome (PCS) and propose practical recommendations concerning safe exercise programming. This is indeed a topic of interest given the increased focus on the importance of effective and safe rehabilitation measures in the large number of people affected by PCS. The manuscript suffers from a superficial review of this idea and a total number of 6 studies, especially only randomized controlled trials (RCTs) were met the inclusion criteria. Moreover, the results suggest exercise training enhance aerobic fitness and physical function, relieve the symptoms of dyspnea, fatigue and depress, and then lower Post COVID-19 Functional Status scores. Unfortunately, the small included literatures, the large differences in the modality, intensity, and frequency of exercise intervention, and the heterogeneity of the outcomes dimmish the overall impact. The paper needs very significant improvement.
This study focuses on the effectivity and safety of home exercise rehabilitation for patients with post-COVID 10 syndrome, which has very good research value and social practice significance.
Limitations:
The beneficial effects vary greatly at different modality, amount and frequency of exercise intervention in the patients with post-COVID-19 syndrome, and there are obvious differences in response to exercise intervention. Moreover, too small included researches make the research results unconvincing.
Firstly, we thank Reviewer 1 for showing interest in our manuscript, recognizing the significance of the topic and drawing close attention to the works limitations. We certainly agree that our review included a relatively small number of studies. With an updated search of the literature, we were able to identify one more study that fits the criteria and that was added to the manuscript. Furthermore, we revised the sections 4.5 Limitations (lines 387-408) and 5. Conclusion (lines 410-435) in order to shed more light on the limitations of the included trials as well as the heterogeneity of outcome measures and exercise modalities that were used in the interventions.
Up to now, there is unfortunately only limited research that examines exercise in PCS patients. The published studies vary greatly in their design and the type of exercise they employ, which is why our review inevitably suffers from these limitations. We nevertheless think, that due to an urgent demand for rehabilitation measures and the ongoing debate about safety concerns when prescribing exercise to PCS patients, this work brings preliminary, yet valuable information to healthcare professionals and directs future research towards more targeted designs.
- What is the time frame for inclusion? From 15 March 2023 to 4 May 2023, is the time of the literature search conducted or included in the publication? Please clarify.
The time frame for inclusion of studies was added as a criterion (Methods line 101) and the literature search was updated (line 93). Furthermore, a slight change in wording in line 91 was made to make it clearer, that the time frame March 15th, 2023 - May 4th, 2023 represents the time where the initial literature search was conducted.
- Should the sentence “Exercise enhanced …… and symptoms of dyspnea …” be replaced as “Exercise enhanced …, and relieved symptoms of dyspnea…”. Please clarify.
Thank you for pointing out this wording mistake, it was changed accordingly in line 19.
- It was suggested that the practical recommendations and discussion in Part 4 should correspond to Part 3, i.e., to make corresponding practical recommendations for the results of the study.
In Part 4, we aimed to make practical recommendations that are in line with both, the results presented in the paper as well as common rehabilitation approaches for similar conditions (section 4.3). Thus, we believed that it would be most suitable to formulate a summarizing synthesis rather than point-by-point comments that correspond to the individual studies.
Reviewer 2 Report
The article presents a review of the literature on the feasibility and efficacy of exercise interventions in patients with Post COVID-19 Syndrome (PCS). I provide a critical commentary on the article below:
Strengths:
Extensive literature search: The article shows that a comprehensive search of relevant medical databases has been conducted, which increases the likelihood of identifying relevant studies on the topic.
Details of included studies: A detailed table is provided summarizing the included studies, including sample, study type, exercise interventions, and outcomes. This facilitates understanding of the different intervention approaches and the results obtained.
Outstanding results: The results of the included studies are presented, showing improvements in functional capacity, symptoms and quality of life of patients with SCA after exercise intervention. These findings are relevant to the medical community and may guide future research and clinical practice.
Points for improvement:
Limitations of the included studies: although the positive results of the studies are highlighted, it is also important to mention the limitations or possible biases in the design and implementation of the studies. This would help readers assess the quality of the evidence presented.
Date of search: The search date mentioned in the article is March 15 to May 4, 2023. Since this is July 2023, the information presented may be outdated at the time of reading. It would be helpful to provide a more current cutoff date to ensure the relevance of the included studies.
Stronger conclusions: The article concludes with practical recommendations based on the literature review. However, it would be beneficial to add a broader discussion of the implications of the results, limitations of the evidence, and areas for future research. This would help contextualize the findings and provide a more complete view of the topic.
The methodological limitations of the included studies, not including only RCTs makes it very difficult to obtain concrete results and valuable conclusions for readers. I believe that the current literature allows a broader approach to this topic, being necessary to further refine the inclusion and exclusion criteria of the articles and thus allowing to reach more concrete conclusions for the readers.
Overall, the article provides valuable information on the efficacy of exercise in patients with SCA and highlights the need for future research to better understand optimal intervention approaches. However, it is suggested to address limitations and provide a more detailed discussion to support conclusions and recommendations. In addition, it is important to keep the information up to date so that the article remains relevant to the medical field.
Author Response
Replies to Reviewer 2:
The article presents a review of the literature on the feasibility and efficacy of exercise interventions in patients with Post COVID-19 Syndrome (PCS). I provide a critical commentary on the article below:
Strengths:
Extensive literature search: The article shows that a comprehensive search of relevant medical databases has been conducted, which increases the likelihood of identifying relevant studies on the topic.
Details of included studies: A detailed table is provided summarizing the included studies, including sample, study type, exercise interventions, and outcomes. This facilitates understanding of the different intervention approaches and the results obtained.
Outstanding results: The results of the included studies are presented, showing improvements in functional capacity, symptoms and quality of life of patients with SCA after exercise intervention. These findings are relevant to the medical community and may guide future research and clinical practice.
Points for improvement:
Limitations of the included studies: although the positive results of the studies are highlighted, it is also important to mention the limitations or possible biases in the design and implementation of the studies. This would help readers assess the quality of the evidence presented.
We thank Reviewer 2 for providing a detailed report and pointing out both the strengths and points of improvement of our manuscript.
We revised section 4.5 (Limitations, lines 387-408) in order to shed more light on the low quality of studies and the possible risk of bias. In addition, we provided more information on the heterogenous modalities used in the studies and the, in part, missing information on exact exercise type and intensity. Hopefully, this will make readers more aware of the limitations of this evidence and guide future directions of research.
Date of search: The search date mentioned in the article is March 15 to May 4, 2023. Since this is July 2023, the information presented may be outdated at the time of reading. It would be helpful to provide a more current cutoff date to ensure the relevance of the included studies.
The search was updates on July 26th (which represents the new cut-off date; line 93). One new study that met the criteria was added to the paper.
Stronger conclusions: The article concludes with practical recommendations based on the literature review. However, it would be beneficial to add a broader discussion of the implications of the results, limitations of the evidence, and areas for future research. This would help contextualize the findings and provide a more complete view of the topic.
We undoubtedly agree with this point of improvement and thus revised the limitation and conclusion sections of the review. Specifically in the final lines (427-433) we tried to draw attention to the limited evidence and the implications for future research.
The methodological limitations of the included studies, not including only RCTs makes it very difficult to obtain concrete results and valuable conclusions for readers. I believe that the current literature allows a broader approach to this topic, being necessary to further refine the inclusion and exclusion criteria of the articles and thus allowing to reach more concrete conclusions for the readers.
Up to this date, a vast majority of studies only included hospitalized patients post COVID-19. With the current criteria that are described in the Methods (i.e. with a focus on non-hospitalized patients), one more relevant study could be identified during the updated literature search. Due to the lack of RCTs, it was not possible to refine the inclusion and exclusion criteria and only include studies with this design, although we agree that this practice would provide the best evidence. However, we highlighted the methodological limitations of the studies and their heterogeneity in the limitations and make readers aware that this also affects the evidence presented in our paper. In light of the urgency of the topic and the great need for rehabilitation measures in PCS, we nevertheless believe, that this narrative review is valuable for readers.
Overall, the article provides valuable information on the efficacy of exercise in patients with SCA and highlights the need for future research to better understand optimal intervention approaches. However, it is suggested to address limitations and provide a more detailed discussion to support conclusions and recommendations. In addition, it is important to keep the information up to date so that the article remains relevant to the medical field.
We are certain that the quality of the manuscript improved by implementing these remarks, i.e. by updating the literature search and adding one more relevant study, pointing out the works limitations in more depth and providing a more detailed discussion of the implementations for future research.
Reviewer 3 Report

Minor editing is needed.
Author Response
Replies to Reviewer 3:
This study is most interesting as there is a growing interest regarding the rehabilitation regimens that could ameliorate the long going post covid symptoms that affect a large number of survivors of the covid pandemic. The authors tried to evaluate the effectiveness of different rehabilitation programs in covid-patients that were not hospitalized. Indeed little attention has been given to them, despite presenting a significant percent. The authors also continued with presenting some suggestions for clinical practice, which is of high importance.
Major concerns:
- Section 3.2. There are a lot of data to be presented, and will be better described or summarized in different tables and not all included in table 1. The exercise intervention is described in the table 1, in text at the section 3.2 in a quite confusing way. I believe that these information will be best presented in a separate table. The readers will more easily find the appropriate information and will be able to compare and critical assess the different programs taking into consideration the huge heterogeneity between them in terms of interventions, time that they were applied and duration.
We thank the reviewer for the thorough analysis of the manuscript and providing such detailed comments that certainly helped us to improve our work.
We presented the data of the included studies in two different tables, as suggested. Table 2 (line 148) now provides all the relevant data about the exercise interventions: type of intervention; duration and frequency, and type and intensity of exercise. Additionally, the same information was reduced in the text (line 133) in order to present the information in a more comprehensive and less repetitive way.
- Section 3.4 presents results of patients reported outcomes. The great number of information being presented should be reduced to the ones that reach statistical significance between groups, in order to highlight the most important outcomes and not all. Table 1 summarizes findings in the best way, so writing should be limited to the most important and those presented in most studies.
As suggested, we removed the information about non-significant results from section 3.4. Only 2 of the included trials included a between-group analysis of the data. We believe that the significant pre-post improvements in the remaining studies would have been looked over if not also presented in the text. However, we limited the written results to the most relevant measures (dyspnea, PCFS score, fatigue, HRQoL) and referred to Table 2 for the rest of the outcomes related to mental well-being. We hope that by making these changes the data is presented in a clearer way.
- Section 3.5 the title of this section is safety and feasibility. These seems to be described in relation to adverse events and to the rate of drop-out, including an assessment prior to the exercise participation. Yet, the authors should present what exactly was assessed during the intervention in relation to safety (was it saturation, heart rate, Borg scale).
In lines 249-255, we added a new paragraph that describes the additional monitoring tools during the interventions.
They only mention that no adverse events were reported. What about discontinues of the session and not a drop-out.
We could not identify any information about the discontinuing of sessions in any of the studies. However, we updated the information about dropouts (when given by the authors of the papers) and added the reasons for dropouts from the interventions (lines 237-239). Furthermore, we draw attention to the issue regarding the partly missing information about adherence and dropouts in lines 417-421.
CPET has been proposed as a tool to best assess the ability of these patients to participate in exercise programs and to tailor the program to their ability and needs. Yet this is poorly described by the authors here and in section 4. 1 or 4.2.
We strongly agree with this point of improvement and recognize the importance of CPET as an assessment tool in PCS patients. A paragraph about CPET and its role when prescribing exercise has been added to section 4.1 (lines 282-286).
In section 4.4 linew 326-327, the authors state that the interventions were well tolerated by presenting data from only 2 studies in relation to heart rate and borg scale.
Heart rate was monitored in three studies, RPE scales were used in four studies. We updated this information in the beginning of section 4.4. Due to the absence of adverse events and no dropouts that occurred due to worsening of symptoms, we believe it can be assumed that the exercise was well tolerated. However, we recognize the concern, especially due to the fact that there is missing data about dropouts or the adherence rates. As pointed out in a previous section, this limitation has been raised in the conclusion (lines 417-421).
Feasibility is among the aims of the study, yet is not well described.
We agree with this major concern and thus removed “feasibility” from section 3.5. and the overall aim of the paper.
Minor concerns:
- The authors state that patients were distinguished according to the severity of their symptoms with the Post-Covid functional scale, but this isn’t included in all studies (4 out of 6- section 3.4/line 184). How patients were categorized in the other two studies?
There is only one study where patients were categorized as such and analyzed in separate groups (Hasenoehrl et al.). In the rest of the studies, there was no categorizations of subjects according to the severity of the condition or other factors.
- Exercise intolerance has a significant place in assessing these patients, yet this is done through a questionnaire. As there isn’t any mention of a more clinical tool, the authors could present any data regarding the reliability of the questionnaire and if there is a correlation to clinical assessments.
Thank you for pointing out this relevant remark. We added information about the reliability and the clinical utility of the DSQ-PEM questionnaire in lines 307-309.
- The patients that were hospitalized, were they admitted in an ICU; how long did they stay at the hospital and during at that period did they have any physiotherapy apart from respiratory/chest?
In Table 1, we added data about the hospitalized patients, when applicable to the study. This information is what we were able to extract from the publications or the supplementary material. Unfortunately, in most cases some of the information (i.e. ICU admission, length of stay) was missing. Furthermore, the authors did not report the application of physio- or other therapies during the hospital stay of patients and we added this concern to the list of limitations (lines 391-393).
The authors have a lot of data to present, although they need to distinguish them in relation to whether they have been included in the majority of studies and there was a significant change from the intervention.
Recommendations are important. The authors should differentiate the terms rehabilitation and exercise in sections 4.3 and 4.4 as there isn’t rehabilitation without exercise.
We recognize that exercise is a key component of rehabilitation. Section 4.3 aims to describe the successful use of rehabilitation programs in similar conditions and identify the (exercise) components that could be similarly beneficial for PCS patients. In section 4.4 we provided more specific recommendations in regards to the most important modalities: endurance and resistance exercise. We hope that by adding an additional, explanatory paragraph at the end of this section (lines 342-346), this is made clearer.
A significant work, that could give valuable information to clinicians, if data presented accordingly.
These comments have undoubtedly improved the quality of our work and we hope that by making the relevant changes to the data presentation, the manuscript results are now displayed in a more comprehensive way.
Reviewer 4 Report
Comments to the Author
The authors of this article did an admirable job on an important topic, aimed to summarize the results of trials investigating exercise interventions in non-hospitalized patients with post-COVID-19 syndrome and propose practical recommendations concerning safe exercise programming. However, there are several points that require further clarity;
Page 2 - Line 29: In this section discussing exercise, it might be relevant to explore the 'open window theory.' Additionally, it is essential to address the impact of COVID-19 infection on professional athletes who engage in regular exercise. Specifically, we should highlight how the infection affects pulmonary function and exercise capacity in elite athletes, and whether persistent symptoms differ from those experienced by the general population. To support these points, here are a few studies for you:
https://doi.org/10.1016/j.resp.2022.103983
http://dx.doi.org/10.1136/bjsports-2020-102789
https://doi.org/10.1007/s11845-021-02849-z
GENERAL COMMENTS:
The study is generally well-written, but it could benefit from the addition of an extra section focusing on the athlete population. Alternatively, a paragraph could be inserted into the relevant section, as previously mentioned.
The language is pretty good maybe it can need to improve a little bit.
Author Response
Replies to Reviewer 4:
The authors of this article did an admirable job on an important topic, aimed to summarize the results of trials investigating exercise interventions in non-hospitalized patients with post-COVID-19 syndrome and propose practical recommendations concerning safe exercise programming. However, there are several points that require further clarity;
We thank the Reviewer for the constructive feedback and pointing out remarks about some missing information in our manuscript.
Page 2 - Line 29: In this section discussing exercise, it might be relevant to explore the 'open window theory.'
After careful revision of the paper, we could not identify a section where we could place information about the open window theory. We believe that it is a highly relevant topic when discussing the acute effects of exercise and perhaps the susceptibility to viral infections. However, this paper aims to explore the chronic effects of exercise in PCS patients. In case we misunderstood this comments’ intention, we are open to receive a more detailed explanation on why the concept could be relevant to our work.
Additionally, it is essential to address the impact of COVID-19 infection on professional athletes who engage in regular exercise. Specifically, we should highlight how the infection affects pulmonary function and exercise capacity in elite athletes, and whether persistent symptoms differ from those experienced by the general population. To support these points, here are a few studies for you:
https://doi.org/10.1016/j.resp.2022.103983
http://dx.doi.org/10.1136/bjsports-2020-102789
https://doi.org/10.1007/s11845-021-02849-z
In lines 54-57, we added information about the prevalence and the symptomatology of athletes affected by long lasting symptoms. We also mentioned, that a decrease in performance could be due to impaired ventilatory patterns. Additionally, we included a section about athletes returning to sports post SARS-CoV-2 infection (lines 270-273). One of the recommended papers was of great use when adding this information and we recognize the importance of not overlooking athletes in this discussion.
Even though the other two suggested publications hold valuable information about the respiratory function of athletes after an infection with SARS-CoV-2, we could not include them as literature in the current review. This is due to the studies not examining samples of athletes who fit the common criteria of PCS (i.e. long-lasting symptoms more than 12 weeks post infection).
GENERAL COMMENTS:
The study is generally well-written, but it could benefit from the addition of an extra section focusing on the athlete population. Alternatively, a paragraph could be inserted into the relevant section, as previously mentioned.
We certainly recognize the importance of including the sports athlete population in the general discussion about PCS and its rehabilitation and hope that we could highlight this by adding two paragraphs to our manuscript. Unfortunately, we did not devote more sections to this topic, since it is not our field of expertise. However, we highly encourage future research being devoted to the multifaceted consequences of COVID-19 in athletes.
Round 2
Reviewer 1 Report
The authors have made some detailed revisions to the manuscript, added an updated article for inclusion, and addressed most of my concerns reasonably.